# Restoration of Joint Line Obliquity May Not Influence Lower Extremity Peak Frontal Plane Moments During Stair Negotiation

**DOI:** 10.3390/bioengineering12080803

**Published:** 2025-07-26

**Authors:** Alexis K. Nelson-Tranum, Marcus C. Ford, Nuanqiu Hou, Douglas W. Powell, Christopher T. Holland, William M. Mihalko

**Affiliations:** 1College of Graduate Health Sciences, The University of Tennessee Health Science Center, Memphis, TN 38163, USA; anelso70@uthsc.edu (A.K.N.-T.); douglas.powell@memphis.edu (D.W.P.); 2Department of Orthopaedic Surgery and Biomedical Engineering, Campbell Clinic, The University of Tennessee Health Science Center, Memphis, TN 38163, USA; marcusfordmd@gmail.com (M.C.F.); cholland@campbellclinic.com (C.T.H.); 3Campbell Clinic Foundation, Germantown, TN 38138, USA; nhou@campbell-foundation.org

**Keywords:** joint line obliquity, restoration, total knee arthroplasty, stair descent, stair ascent, biomechanics

## Abstract

Approximately 15% of total knee arthroplasty (TKA) patients remain dissatisfied after surgery, with joint line obliquity (JLO) potentially affecting patient outcomes. This study investigated whether JLO restoration influenced lower extremity frontal plane joint moments during stair negotiation by TKA patients. Thirty unrestored and twenty-two restored JLO patients participated in this study and were asked to perform five trials on each limb for stair negotiation while three-dimensional kinematics and ground reaction forces were recorded. Frontal plane moments at the ankle, knee and hip were calculated using Visual 3D. The restoration of JLO did not alter frontal plane joint moments during stair negotiation. Both groups showed symmetrical moment profiles, indicating no significant biomechanical differences between the restored and unrestored JLO groups. Restoring JLO did not affect frontal plane joint moments during stair negotiation, suggesting it may not contribute to patient satisfaction disparities post-TKA. Further research should explore other factors, such as surgical technique and implant design, that might influence recovery.

## 1. Introduction

Total knee arthroplasty (TKA) is a treatment for end-stage osteoarthritis (OA) and rheumatoid arthritis patients with pain and functional limits [1]. OA is the most common type of arthritis globally [2,3], and over 50% of those cases affect the knee [4]. With an aging global population, TKA procedures are expected to rise substantially [5,6]. TKA’s primary goals are to relieve pain, improve mobility, and enhance quality of life, but, despite advances in implant design and surgical technique, approximately 15% of patients remain dissatisfied after undergoing TKA [7,8,9,10]. Their satisfaction often hinges on pain reduction and the ability to perform activities of daily living (ADLs).

Knee function is central to ADLs, particularly locomotive tasks such as walking and stair negotiation. The knee is especially susceptible to excessive mechanical loading, with high medial compartment forces linked to OA progression and potential implant loosening. While direct in vivo measurements of these forces are limited, maximum internal knee abduction moments (KAMs) serve as a widely accepted surrogate measure of internal knee loading [11,12]. KAMs result from the interaction between ground reaction forces (GRFs) and their mediolateral distance from the knee joint center of rotation [13]. These moments are associated with medial joint loading, OA severity [14,15,16], and tibial baseplate loosening in TKAs [17,18,19]. During stair negotiation, internal knee loading, including KAMs, is elevated compared to level walking [20,21]. Biomechanical strategies and muscular inputs affect KAM magnitudes [22]; quadriceps weakness, for instance, compromises knee stability and shock attenuation, potentially increasing KAM values [23,24,25]. The sensitivity of knee joint loading to skeletal alignment and muscle function underscores their importance in optimizing TKA outcomes. KAMs also are influenced by distal factors, such as trunk lean and foot adduction angle [26,27]. Additionally, changes in knee loading may impact surrounding lower extremity joint kinematics and kinetics, including the ankle and hip, after TKA [28]. Interpreting moments of all lower extremity joints provides a more comprehensive understanding of biomechanical strategies to perform the task.

Ongoing debate surrounds the optimal restoration of the joint line in TKA. A key parameter in this discussion is joint line obliquity (JLO), defined as the angle between the tibial plateau and the support surface. Mechanical alignment (MA), the traditional TKA approach, aims to restore neutral alignment relative to the mechanical axis, typically 180° ± 3°, to optimize long-term outcomes [29,30,31,32,33,34]. However, MA may not replicate a patient’s natural knee alignment or gait dynamics, potentially impacting patient-reported outcome measures (PROMs). Furthermore, studies show that achieving neutral alignment does not consistently improve implant survival compared with non-neutral alignments [35], and that residual malalignment has minimal impact on PROMs [36,37].

In contrast, kinematic alignment (KA) seeks to restore the patient’s native lower limb alignment and JLO, potentially improving PROMs and knee flexion outcomes [38,39]. KA has also been associated with reductions in KAMs, which may enhance implant longevity [40]. Nonetheless, findings are inconsistent: while some studies highlight significant benefits of KA, others report no measurable improvement in PROMs or joint biomechanics [41,42]. One challenge with KA is accurately restoring the native JLO, given the difficulty in identifying a patient’s original skeletal alignment. Variations in postoperative JLO may contribute to the inconsistent outcomes reported across studies, underscoring the importance of better understanding its biomechanical consequences.

Assessment of knee joint alignment pre- and post-operatively is commonly performed by using long-standing or short knee radiographs. Long-standing lower extremity radiographs, including the femoral head, knee, and ankles, are commonly used by orthopedic surgeons to evaluate mechanical axis alignment and identify deformities [43]. However, there is growing interest in whether single short knee radiographs (SKRs) can provide comparable diagnostic information while reducing radiation exposure and improving cost-effectiveness. Recent literature supports the use of SKRs as a valid alternative, particularly for postoperative assessment [44].

Despite the central role of alignment in TKA, limited research has specifically examined JLO and its effect on patient biomechanics during ADLs. Therefore, the primary aim of this study was to investigate the influence of JLO on frontal plane joint moments in the lower extremity during stair negotiation. Inter-limb symmetry was also analyzed. We hypothesized that patients with similar pre- and post-operative knee joint alignments would demonstrate smaller knee abduction moments compared with those who had differing pre- and post-operative alignments.

## 2. Materials and Methods

### 2.1. Participants

After this study received Institutional Review Board approval, 52 patients who received a cruciate retaining implant were selected and split into two groups based on JLO measures: 30 unrestored and 22 restored. JLO was assessed by one researcher using a weight-bearing anteroposterior (AP) radiograph (Table 1). A perpendicular line to the floor was drawn and a line was drawn across the femoral condyles. The angle between the line across the femoral condyles and the perpendicular line to the floor was measured. JLO was compared between pre-operative and post-operative radiographs to classify groups. The threshold to quantify JLO classifications by the JLO angle was defined as follows: <−3 is apex proximal, −3 < apex neutral < −+3, and apex distal < +3 degrees. A restored joint line was defined as a joint line classification that was maintained in the post-surgery compared with pre-surgery TKA radiographs. An unrestored joint line is defined as the joint line changing JLO classification groups from pre- to post-TKA. All patients met the following inclusion criteria: were one-year post-TKA, between 18 and 80 years old, and ambulated independently. Patients were excluded if they had a bilateral TKA, were prescribed an orthosis to ambulate, or possessed any condition that would limit the patient’s ability to ambulate. Before testing, anthropometric measures were recorded of height and weight (Table 2). Patients were asked to complete two PROMs surveys: knee injury and osteoarthritis outcome score for joint replacement (KOOS, JR) and Forgotten Knee Joint Score (FJS). KOOS, JR measures overall knee health by evaluating stiffness, pain, function, and ADLs for individuals after TKA. FJS measures the patient’s awareness of his or her joint replacement. KOOS, JR focused on the question relating to the studied task, question 4 to evaluate pain going up or down stairs and FJS focused on question 6 to evaluate the awareness of the joint replacement when climbing stairs.

### 2.2. Protocol

Each participant performed 10 stair ascent trials and 10 stair descent trials, initiating five trials on each limb. Participants were provided several minutes of practice before testing to familiarize themselves with the task. Before recorded trials, participants descended the stairs for three trials at a self-selected pace to calculate their average velocity.

The staircase consisted of three steps with a static handrail as referenced by previous work [45]. Participants were instructed to initiate the stair descent task with either their TKA limb or control limb (non-surgical limb) with one foot per stair, with or without using the handrail. Successful trials were performed at a 10% range of their average self-selected velocity (±5%) and with one foot on each stair step. All participants completed five successful trials. An 8-camera markerless motion capture system (200 Hz, OptiTrack, NaturalPoint Inc., Corvallis, OR, USA) paired with AI-based reconstruction software (Version 2023.1.0, Theia Markerless, Inc., Kingston, ON, Canada) was used to collect three-dimensional kinematic data, while GRFs were simultaneously recorded through a force plate-integrated three-step staircase (1000 Hz, AMTI Inc., Watertown, MA, USA). The accuracy and reliability of markerless motion capture systems has been reported in the literature [46,47].

### 2.3. Data Analysis

Kinematic and GRF data were processed using a fourth-order, zero-lag Butterworth filter, with cutoff frequencies set at 10 Hz for kinematics and 50 Hz for GRFs. Stair locomotion biomechanics were analyzed from initial contact (IC) to toe off (TO) based on previous literature [48]. IC was defined as the instant from which the vertical GRF exceeded 50 Newton (N) for a period greater than 20 milliseconds (ms). TO was defined as the point after IC in which the vertical GRF fell below 50 N for at least 20 ms. Joint moments were normalized to patients’ body mass.

Visual3D (HAS-Motion, Inc., Kingston, ON, Canada) was used to calculate frontal plane lower extremity moments. Peak frontal plane joint moments were defined as the maximum value of the joint moment during IC to TO and extracted using custom software (Version R2023b, MATLAB, Natick, MA, USA). Subject means were calculated as the average of the five trials from each individual during each task (stair ascent and descent, independently).

### 2.4. Statistical Analysis

Data were tested for normality using the Shapiro–Wilk test. Four independent sample *t*-tests were conducted to determine the effect of the restoration of JLO (restored and unrestored) during each task (stair ascent and descent) on peak frontal plane lower extremity joint moments. Wilcoxon-Signed Rank tests were conducted on two non-parametric variables to compare between the JLO groups during stair ascent, hip and knee abduction moments. Similarly, four independent sample *t*-tests were conducted between limbs to determine the effect of the TKA (TKA vs. CON) during each task (stair ascent and descent) on peak frontal plane lower extremity joint moments. Wilcoxon-Signed Rank tests were conducted on two non-parametric variables to compare between limbs during stair ascent, ankle inversion moments, and stair descent, hip abduction moments. Alpha was set at *p* < 0.05. SAS 9.5 software (Version 9.5, SAS Institute Inc., Cary, NC, USA) was used to perform all statistical analyses.

## 3. Results

Table 1 presents anthropometric and patient-reported outcome measures. An unrestored knee joint alignment was observed in 30 patients. Twenty-two patients restored their natural alignment postoperatively. There were no significant differences in age (unrestored = 61.8 ± 7.6, restored = 61.9 ± 7.09, *p* = 0.963), body mass index (BMI) between groups (unrestored = 31 ± 6, restored = 31 ± 6, *p* = 0.500) nor PROMs were significantly different between groups (KOOS JR, unrestored = 4.1 ± 0.9, restored = 4.0 ± 0.8, *p* = 0.228; FJS, unrestored = 3 ± 1.5, restored = 3 ± 1.5, *p* = 0.642).

### 3.1. Stair Ascent Biomechanics

Table 3 and Figure 1 present ankle, knee, and hip peak moments comparing limb (TKA vs. CON) and group (unrestored vs. restored) during the stair ascent task. No significant effect of TKA nor JLO was observed for peak ankle inversion (TKA: *p* = 0.170; JLO: *p* = 0.742), knee abduction (TKA: *p* = 0.086; JLO: *p* = 0.162), nor hip abduction (TKA: *p* = 0.114; JLO: *p* = 0.745) joint moments during stair ascent.

### 3.2. Stair Descent Biomechanics

No significant effect of TKA nor JLO was observed for peak ankle inversion (TKA: *p* = 0.987; JLO: *p* = 0.143), knee abduction (TKA: *p* = 0.294; JLO: *p* = 0.745), or hip abduction (TKA: *p* = 0.934; JLO: *p* = 0.500) joint moments during stair descent (Table 3, Figure 1).

## 4. Discussion

This study found that JLO did not influence lower extremity frontal plane joint moments during stair negotiation, and that patients were fully recovered between limbs and did not show frontal plane asymmetries to negotiate the stairs. While previous literature has investigated the effects of JLO on frontal plane knee joint moments during overground walking, this was the first investigation on JLO’s influence on lower extremity biomechanics during stair negotiation [42].

Limited research has investigated restoration of knee JLO on stair negation biomechanics; however, our findings can be compared with previous literature examining a similar technique to restore joint line through the KA technique. There is avid debate within the field of orthopedic surgery on the safe alignment limits of the knee joint to the kinematic axes. Knee joint alignment beyond 3° of the mechanical axis has been shown to increase the prevalence of aseptic loosening, and implant wear rates due to the risk of excessive loading of the medial compartment of the knee prosthesis [44,45]. Despite the increased mechanical demand induced by stair negotiation as well as the differences in JLO, KAMs remained unchanged. These findings suggested that compensatory mechanisms could be observed in other planes of motion (sagittal or transverse), at adjacent joints on the ipsilateral limb or within the contralateral limb to avoid high loads within the medial compartment of the surgical knee.

The unique lower extremity kinetic patterns in restored and unrestored JLO TKA patients represent the interaction of the mechanical forces, lower extremity structural alignment and neuromuscular activation patterns. If knee joint kinetics are influenced by JLO, changes to the biomechanical strategies at the ipsilateral ankle and hip joint kinetics, and contralateral limb kinetics could occur. Both JLO groups displayed inter-limb symmetry as well as no differences in lower extremity moment profiles (in the frontal plane) during stair negotiation. Symmetry between limbs and in surrounding joints purportedly acts to prevent progression of OA by limiting medial compartment loading [17,49]. Our symmetry findings were similar to previous literature that demonstrated no differences in frontal plane lower extremity moment profiles between limbs at one-year post-TKA [50,51]. There is limited research in the literature that compares the influence of TKA on lower extremity biomechanical profiles beyond the knee joint (surgical compared to non-surgical). A study by Standifird et al., reported asymmetrical frontal plane moment profiles in the knee and hip joints as compared with the contralateral limb in patients post-TKA during a stair ascent and descent task [52]. However, these previously reported findings had limited control on the post-surgical timeline and included patients 6 months post-TKA. In the current study, all patients had to be more than 12 months post-TKA, suggesting they fully recovered and rehabilitated from the TKA surgery. A number of research studies have demonstrated that patients continue to exhibit asymmetrical locomotor patterns at 6 months post-TKA [53,54]. These methodological differences in patient inclusion criteria could result in the distinct symmetry findings reported by the two research studies.

## 5. Limitations

This valuable study still has some limitations. It is known that trunk lean could alter KAMs, during stair ascent and descent. A post hoc analysis of trunk lean in study participants revealed no significant differences in trunk lean between the JLO groups, suggesting trunk lean did not affect KAMs during stair ascent (unrestored = 3.63 ± 2.04, restored 4.64 ± 2.40, *p* = 0.99) nor descent (unrestored = 3.70 ± 2.24, restored 3.76 ± 2.55, *p* = 0.23). Additionally, some participants used the handrail during stair negotiation, which may have influenced trunk and lower extremity biomechanics. However, a sub-analysis excluding handrail users did not alter the results, suggesting that handrail use may not have impacted the findings. JLO was measured using SKR, which also were used to classify patients into restored and unrestored groups. While this method introduces a potential limitation compared with long-standing radiographs, previous studies have demonstrated a strong correlation between short and full-length radiographs, supporting the reliability of SKR for assessing coronal plane knee alignment [44].

Finally, the older age of our study population compared with previous research increased the risk of OA in the non-surgical limb. Therefore, the non-surgical limb in our study was used as a within-subject comparison rather than a “healthy control,” providing a more relevant measure of the surgical intervention’s impact on individual outcomes. These limitations should be considered when interpreting the study’s findings and their implications for treatment and rehabilitation.

## 6. Conclusions

This study found that restoring JLO did not affect frontal plane joint moments. Additionally, more than one year after a TKA, patients exhibited symmetrical lower extremity frontal plane moments, indicating full recovery. Therefore, JLO restoration may not be a key factor in patient satisfaction disparities post-surgery. However, this research had limitations since neither the surgical technique nor the prosthesis was controlled between JLO groups. Future research should explore how surgical alignment techniques and implant designs impact stair negotiation biomechanics in TKA.

## Figures and Tables

**Figure 1 bioengineering-12-00803-f001:**
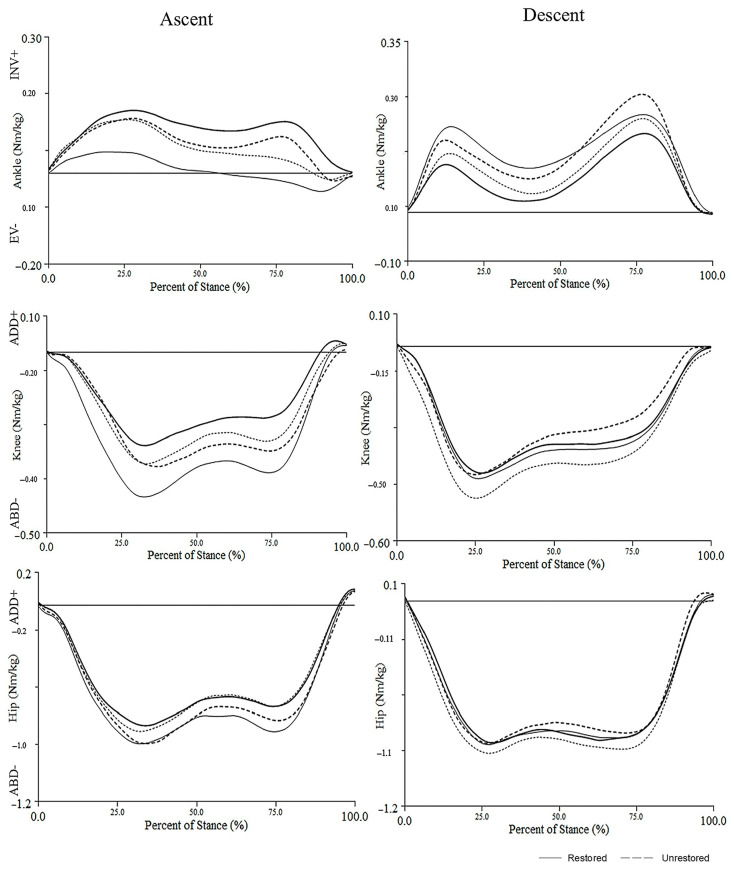
Frontal plane lower extremity moment curves during stair negotiation comparing between joint line obliquity groups and between limbs. Notes: Ensemble averages for full moment curves between joint line obliquity groups represented by bolded lines for the TKA limb (solid line, restored; dashed line, unrestored) and respective control limbs represented by unbolded lines.

**Table 1 bioengineering-12-00803-t001:** Joint line obliquity classification.

Pre-—Post-Op JLO	Restored, *n* (%)	Pre-—Post-Op JLO	Unrestored, *n* (%)
Neutral–Neutral	18 (81.8)	Apex Distal–Neutral	18 (60)
Apex Distal–Apex Distal	3 (13.6)	Apex Distal–Apex Proximal	4 (13.3)
Apex Proximal–Apex Proximal	1 (4.5)	Neutral–Apex Distal	2 (6.7)
		Apex Proximal–Neutral	3 (10)
		Neutral–Apex Proximal	3 (10)

**Table 2 bioengineering-12-00803-t002:** Participant anthropometric measurements in each joint line obliquity group.

Group	*n*	Age	BMI	Male	Female	KOOS, JR	FJS	Years Post-Op
Restored	22	61.9 ±7.1	31 ±6	14	8	4.1 ±0.9	3 ±1.5	2.3 ±1.0
Unrestored	30	61.8 ±7.6	31 ±5	16	14	4.0 ±0.8	3 ±1.5	2.7 ±1.2

Notes: Mean ± standard deviation; BMI, body mass index; FJS, forgotten joint score; KOOS, JR, knee injury, and osteoarthritis outcome score for joint replacement.

**Table 3 bioengineering-12-00803-t003:** Peak lower extremity frontal plane moments between limbs and experimental JLO groups during a stair ascent and descent task.

		Ankle Inversion (Nm/kg)		Knee Abduction (Nm/kg)		Hip Abduction (Nm/kg)	
Task	Limb	Un	Restored	Un	Restored	Un	Restored
Ascent	TKA	0.16 ± 0.13	0.18 ± 0.13	0.40 ± 0.21	0.32 ± 0.13	0.94 ± 0.27	0.84 ± 0.14
	CON	0.14 ± 0.10	0.11 ± 0.11	0.42 ± 0.19	0.45 ± 0.19	0.92 ± 0.21	0.97 ± 0.22
Descent	TKA	0.29 ± 0.16	0.23 ± 0.14	0.50 ± 0.18	0.48 ± 0.22	1.06 ± 0.14	1.03 ± 0.22
	CON	0.26 ± 0.17	0.27 ± 0.13	0.55 ± 0.21	0.51 ± 0.18	1.06 ± 0.28	1.02 ± 0.17

Notes: Mean ± standard deviation; CON, controlled limb; JLO, joint line obliquity,; Un, unrestored.

## Data Availability

The raw data supporting the conclusions of this article will be made available by the authors upon request.

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
