# Peer review of "Restoration of Joint Line Obliquity May Not Influence Lower Extremity Peak Frontal Plane Moments During Stair Negotiation"

_bioengineering, 2025, doi:10.3390/bioengineering12080803_

Round 1

Reviewer 1 Report

Comments and Suggestions for Authors

I found your manuscript titled “Restoration of Joint Line Obliquity Does Not Influence Lower Extremity Peak Frontal Plane Moments During Stair Negotiation” informative and well written. The work is novel, addresses an interesting point and is potentially impactful as well, highlighting that restoration of knee joint line obliquity during total knee arthroplasty does not influence frontal plane joint loading. However, I do have some points for you to consider though as detailed below.

Overarching

  • Do you have preoperative gait data for the patients? Exploring whether restoration of the JLO influences the change in peak moments might be useful for supporting some of the statements around recovery. It could be that restoring the JLO provide greater magnitude of recovery when compared to preoperative values.
  • The work appears based on the premise, and it is not the only study to do so, that restoring symmetry between operated and non-operated limbs is beneficial; however, if both limbs are abnormal in their loading patterns is restoration of this abnormal loading actually a good thing? Additionally, the analysis ran, at least to my understanding, compares TKA to CON but not TKA to CON within each group, the magnitude of change appears to differ quite a lot (at least visually) between groups for KAM during stair ascent. Actually quantifying the symmetry between limbs and comparing between groups may yield a different finding to that suggested currently.
  • Is it possible to explore the data within apex groups or changes across these? I am curious whether changes are inconsistent across the different apex groups and in some senses cancel each other out within the group level analysis currently undertaken? I wouldn’t expect a change from apex neutral to apex distal or neutral to proximal to influence the KAM in the same manner for instance

Introduction

  • Could you potentially add in a sentence or two within the introduction to rationalise the inclusion of hip and ankle moments? I think it is really good you have done so, but highlighting that changes at the knee may impact changes above and below within the kinetic chain would strengthen the rationale for the study further
  • Were no hypotheses proposed in relation to the hip and ankle joint? At the minute the hypothesis is solely related to the knee which in some ways, especially without any rationalisation of the hip and ankle assessments, suggests only the knee should or would be explored
  • Could you also add in the final paragraph of the introduction about the comparison across limbs

Materials and Methods

  • Was a consistent implant utilised across the group?
  • What was the split of the apex groups across the restored and not restored classifications?
  • Was testing undertaken with shoes or barefoot? If shod, then were these standardised in any manner? Differences in footwear may alter frontal plane moments masking the effect of JLO restoration
  • What clothing was worn by participants during testing? Given the use of markerless systems this may influence accuracy across sessions. Additionally, is there any evidence (yours or other publications) you could add in to highlight the reliability of the Theia system in relation to frontal plane assessments? This might help to strengthen the readers confidence in the findings further.
  • In terms of the staircase, was a force plate located in each step or just one?
  • Please add in the Data Analysis section that joint moments were normalised to body mass?
  • Were joint moments calculated within Visual 3D and then peak extracted within MatLab? At the minute it suggestions that both sets of software were used to calculate moments

Results

  • Add a link to figure 1 in within the stair descent section as well, appreciate it is in the accent section but this only refers to ascent not descent, to highlight where kinetic waveforms are

Discussion

  • See overarching comment in relation to the point that no frontal plane asymmetries are evident
  • Flip the order of information in the opening sentence of the second paragraph, from “Limited research has investigated stair negation biomechanics on the restoration of knee JLO” to “Limited research has investigated restoration of knee JLO upon stair negation biomechanics”.
  • Throughout the discussion you appear to be trying to maintain an argument that restoration of JLO is important, but your findings seem to suggest otherwise at least in the frontal plane. Do your findings suggest that restoration of the JLO isn’t as impactful as previously thought? Is it that only very extreme deviations are likely to cause identifiable alterations?
  • What is the relevance of the PROMS sentence in the second paragraph of the discussion? No real discussion of this point it just jumps in and disappears. Do PROMS and biomechanics measure the same thing? There seems to be quite a lot of evidence suggesting limited agreement between the measures.
  • In the third paragraph, when you say “unique” what is the reference point?

Reviewer 2 Report

Comments and Suggestions for Authors

Introduction

Lines 32-33: “Total knee arthroplasty (TKA) is a paramount treatment for end-stage osteoarthritis 32 (OA) and rheumatoid arthritis”’

While I agree that TKA is the most common treatment for end stage OA I would not argue that is paramount as some with KL-4 do not seek TKA. I would either temper the language and remove the word paramount or keep it and add that TKA is paramount for end stage patients with pain and functional limitations.

Line 52: “KAMs also are influenced by remote factors, such as trunk lean and foot adduction”

I would revise to say distal factors instead of remote factors to improve readability.

Methods

Line 101: “were one-year post-TKA”

Were all patients exactly one year post TKA or a minimum of one year post TKA? If a minimum of one year post TKA was the criteria average time since TKA would be helpful to include in the demographics section.

Lines 121-123: “Participants were instructed to initiate the stair descent task with either their 121 TKA limb or control limb (non-surgical limb) with one foot per stair, with or without using 122 the handrail”

How do you think allowing the participants to use the handrail may have impacted your results? Was a light touch allowed or were they allowed to fully grip and put weight into the rails?  If the latter this may have impacted some of the kinetic results shown.

Lines 144-146: “Four independent sample t-tests were conducted to determine the effect of the restoration of JLO (restored and un-restored) during each task (stair ascent and descent) on peak frontal plane lower extremity joint moments.”

The stats are hard to follow as written as written it sounds as if you only need 2 t-tests a between group analysis for stair ascent and one for descent are you also looking between limbs, if so that is not clear in the methods as written.  If I am interpreting this correctly you ran a t-test between restored and unrestored groups for each limb on frontal plane mechanics during each task which is not immediately clear as written. Also, why not do 2 ANOVAs for your statistics instead of 4  t-tests? This would preserve your statistical power and reduce the chance of encountering a type 1 error you could do a 2 (restored and unrestored) by 2 (limb) ANOVA for ascent and descent.

Lines 149-150: “four independent sample t-tests were conducted between limbs to 149 determine the effect of the TKA (TKA vs. CON) during each task (stair ascent and descent)”

Same comment as above also why not run a 2 (group) by 2 (condition) ANOVA for each limb?

Round 2

Reviewer 2 Report

Comments and Suggestions for Authors

I commend the authors on their through job addressing all revisions. I believe this manuscript is now suitable for publication.